# Circuit Techniques in GaN Technology for High-Temperature Environments

**Ahmad Hassan** [1,*], **Jean-Paul Noël** [2], **Yvon Savaria** [1] **and Mohamad Sawan** [1,3]

1 Department of Electrical Engineering, Polytechnique Montréal, Université de Montréal, Montréal, QC H3T 1J4, Canada; yvon.savaria@polymtl.ca (Y.S.); mohamad.sawan@polymtl.ca or sawan@westlake.edu.cn (M.S.)

2 National Research Council Canada (NRC), Ottawa, ON K1A 0R6, Canada; Jean-Paul.Noel@nrc-cnrc.gc.ca

3 The Institute for Advanced Study, Westlake Institute for Advanced Study, School of Engineering, Westlake University, Hangzhou 310024, China

\* Correspondence: ahmad.hassan@polymtl.ca

**Abstract:** As a wide bandgap semiconductor, Gallium Nitride (GaN) device proves itself as a suitable candidate to implement high temperature (HT) integrated circuits. GaN500 is a technology available from the National Research Council of Canada to serve RF applications. However, this technology has the potential to boost HT electronics to higher ranges of operating temperatures and to higher levels of integration. This paper summarizes the outcome of five years of research investigating the implementation of GaN500-based circuits to support HT applications such as aerospace missions and deep earth drilling. More than 15 integrated circuits were implemented and tested. We performed the HT characterization of passive elements integrated in GaN500 including resistors, capacitors, and inductors up to 600 °C. Moreover, we developed for the first time several digital circuits based on GaN500 technology, including logic gates (NOT, NAND, NOR), ring oscillators, D Flip-Flop, Delay circuits, and voltage reference circuits. The tested circuits are fabricated on a 4 mm × 4 mm chip to validate their functionality over a wide range of temperatures. The logic gates show functionality at HT over 400 °C, while the voltage reference circuits remain stable up to 550 °C.

**Keywords:** GaN technology; high-temperature electronics; harsh environments; digital circuits; high-temperature characterizations

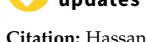



## 1. Introduction

Harsh environment electronics has become a widespread research topic in recent years [1] due to three main factors: (1) the lack of conventional silicon operating at high-voltage and extreme temperature conditions, (2) the evolution of substrate materials that can handle some of the extreme environments found in industrial fields, and (3) the need for more reliable and durable microelectronic devices that can serve industrial harsh environments. In many industrial fields, such as combustion engines, hybrid vehicles, aerospace, and deep earth drilling, there is a need for electronics that operate at extreme temperatures and that can endure high voltages [2]. Although silicon is widely used to implement conventional integrated circuits (ICs) that are suitable for the majority of applications, its use in harsh applications is confined by its maximum operating temperature (<300 °C in case of silicon on insulator (SOI)) [3].

The development of advanced substrate materials, like gallium arsenide (GaAs), silicon germanium (SiGe), silicon carbide (SiC) and gallium nitride (GaN), accelerates the proliferation of ICs usage in harsh environmental applications. Particularly, wide bandgap (WBG) semiconductors, SiC and GaN, are well adapted to withstand high-system voltages and extreme temperatures [4,5]. Those two materials possess a wide bandgap (more than 3 eV), almost three times wider than that of Si (1.1 eV), making the leakage current in WBG devices considerably lower at extremely high temperatures. In addition, the electric

breakdown field of WBG devices is six times larger than that of silicon, allowing them to endure much higher operating voltages.

While GaN is considered to be an excellent choice for high-voltage and high-frequency industrial applications, SiC still has three times the thermal conductivity of Si or GaN. This makes the dissipation of induced self-heating much easier in the implemented circuits/systems [6]. In fact, in recent years, GaN Heterojunction Field Effect Transistors (HFETs) have been extensively studied for high-frequency and power applications. However, the same material properties of the GaN HFET heterostructure that nominate it as a favorable candidate for low-noise and high-power applications, particularly the high breakdown field, wide bandgap, and high saturation velocity, make GaN devices very promising for high-temperature electronics [4]. Combining the excellent thermal conductivity of SiC and the exceptional properties of GaN heterostructure technology, GaN semiconductors processed on SiC substrates are strong candidates for implementing highly durable and stable ICs for extreme ambient temperatures of 500 °C or higher.

However, despite the development and characterization of several GaN devices operating at extreme temperatures exceeding 400 °C [7], 600 °C [8] and 800 °C [9], the implemented ICs based on these devices are very limited with regard to circuit complexity and achieved HT. This includes oscillator, inverter, and level shifter circuits with maximum operating temperatures up to 300 °C [10–14].

GaN500 is a technology fabricated at the National Research Council (NRC) of Canada. It is processed on three inches SiC 75 μm thick wafers. The technology features double fingers 0.5 μm metal gates, nichrome (NiCr) resistors, MIM capacitors and two metal layers for interconnect. The Process Design Kit (PDK) for the technology was originally developed for RF power amplifiers applications as the transistors it offers have fmax as high as 40 GHz, with reliable operation between room temperature (RT) and 200 °C ambient. However, prior to our work, high temperature characterization had not been performed for this technology and it had not been used to implement analog and digital ICs beyond RF power amplifiers.

In this paper, we present the high temperature characterization of passive elements integrated in GaN500 technology up to 600 °C. In addition, we demonstrate the functionality of digital circuits made with GaN500 HFETs over a wide range of temperatures. This development is intended to exploit the inherent ability of GaN500 technology to implement complex ICs and systems comprising modules such as ADC, DAC, modulator, and demodulator operating at temperatures over 300 °C. Section 2 presents the HT characterization results of passive elements, while Section 3 describes the design of different digital circuits based on GaN500 along with their experimental validation over a wide range of temperatures. Finally, the conclusion in Section 4 summarizes the main contributions of this work.

## 2. High Temperature Characterization and Modeling of Passive Elements

Recently, we had investigated the possibility of using GaN500 HEMT technology to implement HT ICs [15–19] at the Polystim Laboratory [20]. The results reported in [15] confirmed the stability of tested GaN devices from room temperature up to 400 °C. In [16], basic digital circuits were implemented using GaN500 and off-chip resistors achieving HT operation up to 400 °C. In addition, three demodulators were demonstrated to recover data from an LSK-based modulated signal. The functionality of the demodulators was validated by simulations. The modulation/demodulation system proposed in [17] was the first GaN500-based integrated data transmission system. This modulation system was based on a simplified delta-sigma modulation technique, and a fully digital demodulator was implemented to recover data from a modulated signal. The reported simulation results confirmed the functionality of the proposed systems over temperatures ranging from 25 °C to 350 °C. The digital demodulator was experimentally validated in [18] at 160 °C. In [19], we performed the HT characterization and modeling of GaN500 devices showing a stable operation up to 600 °C. The characterization results were used to extract

an extended version of the Angelov model of GaN HFETs after considering the temperature as a variable parameter.

Due to the absence of complementary normally-off devices in GaN500 technology, the solution adopted to develop GaN ICs is to use integrated resistors as loads. To ensure an appropriate operation of the developed IC over a wide temperature range, it was proven in [2] that temperature matching between diverse components can be successfully accomplished using integrated resistors. Thus, to obtain the required accurate models that consider the temperature impact on the values of passive elements, we performed an HT characterization of integrated resistors, capacitors, and inductors.

Different sizes of passive components were fabricated, including six coils, five capacitors and nine resistors. The micrographs of these devices are shown in Figure 1, and their values at room temperature are summarized in Table 1. To perform the HT measurements, we used a furnace that can reach 1000 °C. The chamber of the furnace has a top opening that allows to access the tested chip by HT cables. The test chip was wire bonded to a ceramic package that can endure the HT inside the furnace. The measurement setup is shown in Figure 2 and more details about the HT packaging and testing setup can be found in [18].

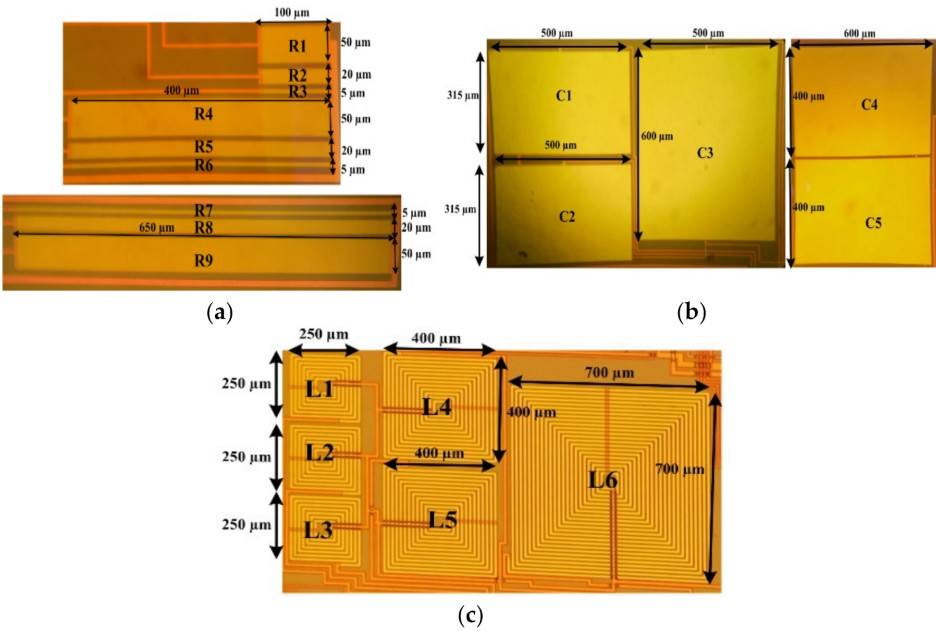

**Figure 1.** Integrated passive components: (**a**) resistors, (**b**) capacitors, and (**c**) inductors.

**Table 1.** Integrated passive elements in GaN500.

| Resistor | Value @ 25 °C (Ω) | Resistor | Value @ 25 °C (Ω) | Capacitor | Value @ 25 °C (pF) | Inductor | Value @ 25 °C (μH) |
|---|---|---|---|---|---|---|---|
| $R_1$ | 102 | $R_7$ | 5160 | $C_1$ | 58 | $L_1$ | 0.99 |
| $R_2$ | 220 | $R_8$ | 1316 | $C_2$ | 56 | $L_2$ | 0.97 |
| $R_3$ | 802 | $R_9$ | 540 | $C_3$ | 83 | $L_3$ | 0.86 |
| $R_4$ | 340 | - | - | $C_4$ | 75 | $L_4$ | 1.5 |
| $R_5$ | 816 | - | - | $C_5$ | 73 | $L_5$ | 1.45 |
| $R_6$ | 3610 | - | - | - | - | $L_6$ | 1.7 |

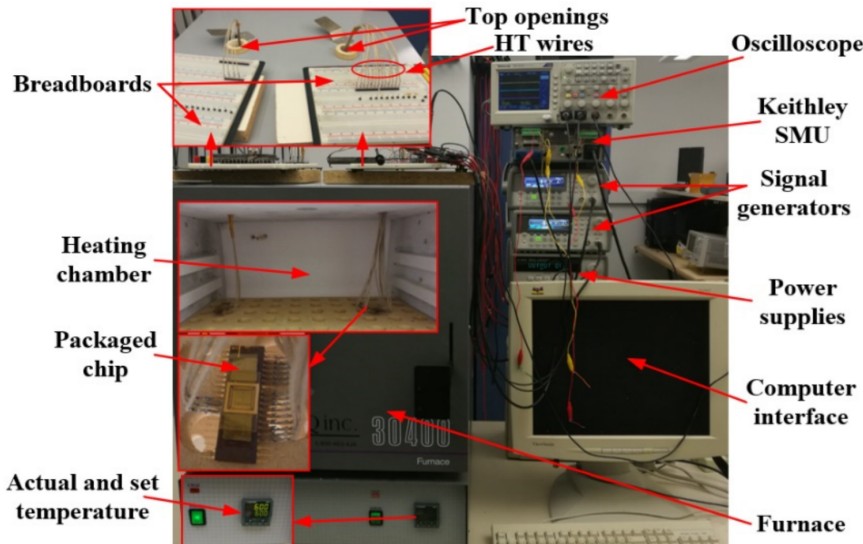

**Figure 2.** Experimental setup used for HT measurements.

### 2.1. Integrated Resistors

The experimental setup shown in Figure 2 is used in this paper to perform the HT characterizations of integrated resistors between 25 °C and 600 °C. It was found that the measured resistances of $R_{1-9}$ linearly increase with temperature in the 25–600 °C range. However, the resistance to temperature slope listed in Table 2 is different from one resistor to another. Figure 3 shows that the relationship between the slope and the initial value of the resistors is linear.

**Table 2.** Experimental characterization results of integrated resistors implemented with GaN500.

| Resistor | @ 25 °C (Ω) | @ 100 °C (Ω) | @ 200 °C (Ω) | @ 300 °C (Ω) | @ 350 °C (Ω) | @ 400 °C (Ω) | @ 450 °C (Ω) | @ 500 °C (Ω) | @ 550 °C (Ω) | @ 600 °C (Ω) | Slope (Ω/°C) |
|---|---|---|---|---|---|---|---|---|---|---|---|
| $R_1$ | 102 | 108 | 118 | 128 | 134 | 140 | 148 | 157 | 164 | 176 | 0.11 |
| $R_2$ | 220 | 230 | 243 | 258 | 266 | 274 | 283 | 296 | 305 | 326 | 0.16 |
| $R_3$ | 802 | 824 | 858 | 893 | 911 | 928 | 944 | 976 | 994 | 1039 | 0.36 |
| $R_4$ | 340 | 353 | 371 | 391 | 400 | 410 | 422 | 440 | 450 | 460 | 0.2 |
| $R_5$ | 816 | 839 | 872 | 909 | 928 | 946 | 967 | 997 | 1008 | 1034 | 0.37 |
| $R_6$ | 3610 | 3240 | 3350 | 3470 | 3530 | 3580 | 3640 | 3720 | 3740 | 3780 | 1.1 |
| $R_7$ | 5160 | 5270 | 5450 | 5640 | 5730 | 5810 | 5910 | 6010 | 6020 | 6050 | 1.6 |
| $R_8$ | 1316 | 1350 | 1400 | 1454 | 1481 | 1505 | 1538 | 1578 | 1591 | 1622 | 0.52 |
| $R_9$ | 540 | 557 | 581 | 607 | 621 | 633 | 651 | 671 | 680 | 698 | 0.26 |

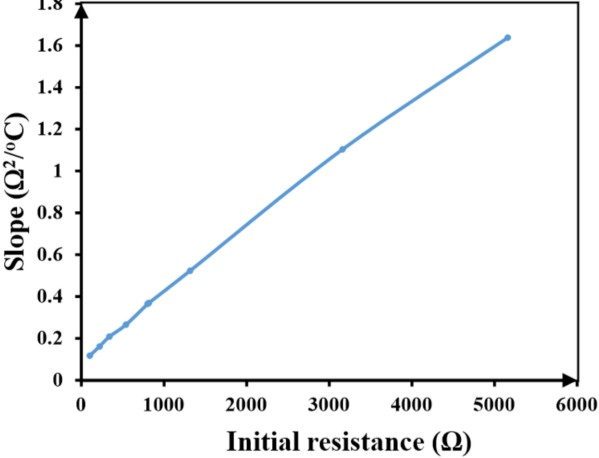

**Figure 3.** Variation of the resistance slope with respect to the initial resistance value.

Equation (1) proposes a general expression for the resistance value as a function of the applied temperature. In (1), $R_0$ is the initial resistance at 0 °C, T is the actual temperature (°C) and $\alpha$ is the slope of the resistance variation with temperature. From Figure 3, we extract the expression for $\alpha$ (Equation (2)) and substitute it into equation (1) to formulate the final expression for the resistance as a function of temperature (Equation (3)). To validate Equation (3), the simulations of $R_{1-9}$ are compared with experimental measurements over the temperature range 25–600 °C. Figure 4 shows the good matching between simulated and measured resistances of the $R_5$, $R_6$, $R_8$, and $R_9$ resistors.

$$R_t = R_0 + \alpha T, \tag{1}$$

$$\alpha = 3.5 \times 10^{-4} R_0 + 0.08, \tag{2}$$

$$R_t = R_0 \left(1 + 3.5 \times 10^{-4} T\right) + 0.08T, \tag{3}$$

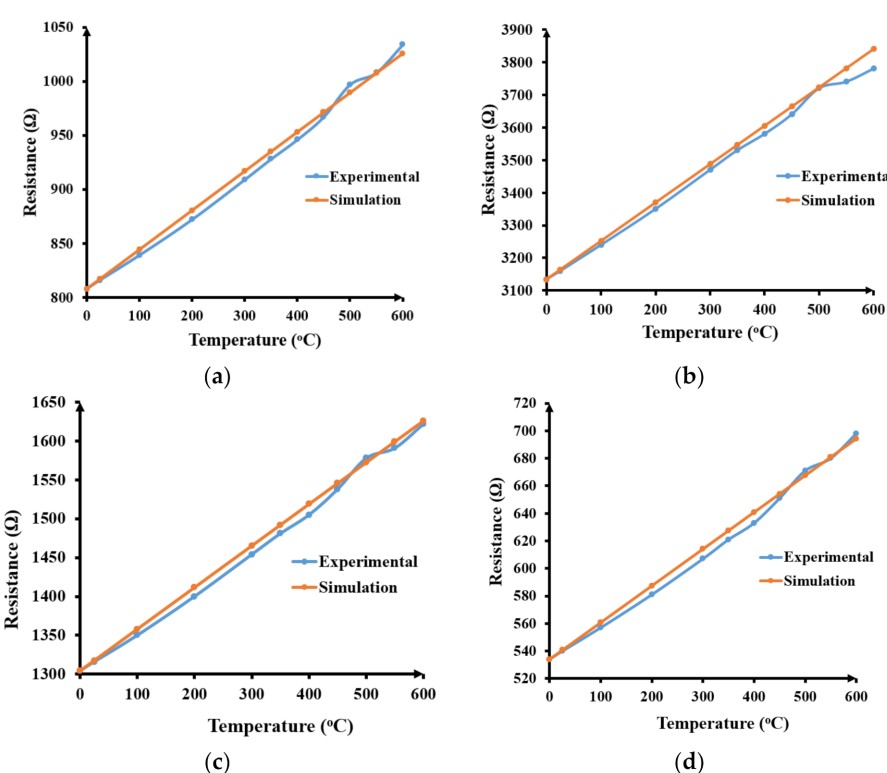

**Figure 4.** Integrated resistors characterization at HT: (**a**) $R_5$, (**b**) $R_6$, (**c**) $R_8$, and (**d**) $R_9$.

## 2.2. Integrated Capacitors

Using an Agilent impedance analyzer, the sensitivity of various capacitor values was characterized, and Table 3 summarizes the values at different temperatures. Figure 5 shows the capacitances of three different capacitors ($C_1 = C_2$ and $C_4 = C_5$) over temperatures ranging between 25 °C and 600 °C. The results show the capacitance stability of each capacitor up to 500 °C, with minor variations due to the variable parasitic capacitance of testing probes. Above 500 °C, the capacitance values started to increase gradually. Therefore, the capacitance could be considered as having a constant value within the temperature range 25–500 °C. Further measurements should be performed at higher temperature to extract temperature models of capacitors covering the temperatures beyond 500 °C.

**Table 3.** Integrated capacitors parametric stability at HT.

| Capacitor | @ 25 °C (pF) | @ 100 °C (pF) | @ 200 °C (pF) | @ 300 °C (pF) | @ 350 °C (pF) | @ 400 °C (pF) | @ 450 °C (pF) | @ 500 °C (pF) | @ 550 °C (pF) | @ 600 °C (pF) |
|---|---|---|---|---|---|---|---|---|---|---|
| $C_1$ | 58 | 59 | 59 | 59.3 | 58.4 | 58.8 | 58.87 | 60.1 | 60.7 | 63.3 |
| $C_2$ | 56 | 55.5 | 55.8 | 56.4 | 58.4 | 56.8 | 56.82 | 57 | 58.3 | 62.1 |
| $C_3$ | 83 | 84 | 84 | 84 | 83.6 | 85.2 | 85 | 86.1 | 86.4 | 89.1 |
| $C_4$ | 75 | 75 | 75.5 | 76.1 | 76.3 | 76.7 | 76.5 | 77.9 | 78.5 | 80.2 |
| $C_5$ | 73 | 73 | 74.5 | 74.6 | 75 | 76.2 | 75.8 | 76.79 | 77.4 | 79.8 |

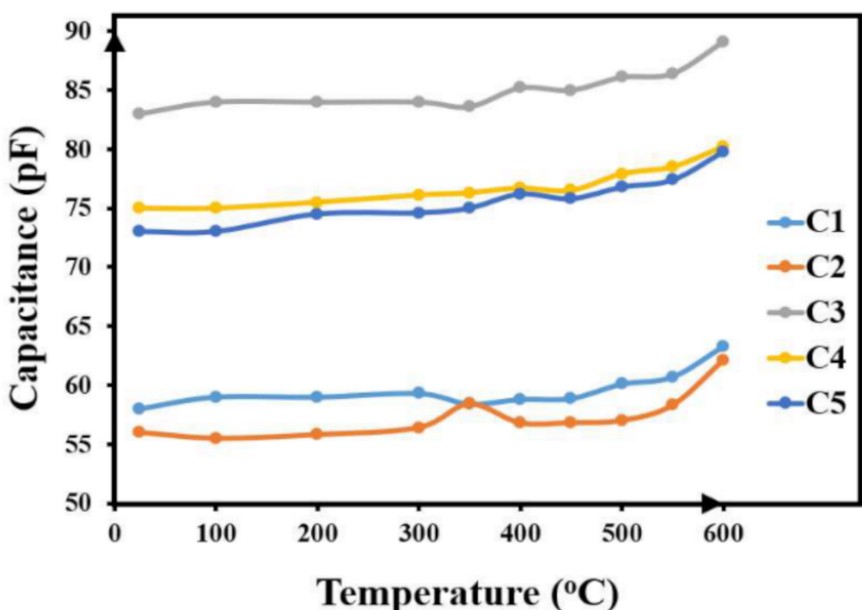

**Figure 5.** Parametric characterization at HT of different integrated capacitors.

### 2.3. Integrated Inductors

The parametric values of all fabricated coils were characterized, and Table 4 summarizes the measured inductances at different temperatures ranging from 25 °C to 300 °C, and Figure 6a plots the measured inductance values showing that they are relatively stable at temperatures up to 200 °C, then these inductances increase gradually with temperature. The small inductance values of the integrated coils are perturbed by the parasitics of the test breadboards as can be seen in Figure 6. The resistances of coils at various temperatures are summarized in Table 5, and Figure 6b shows the growth of these resistances with temperature. We did not pursue further investigation on the integrated coils because our implemented circuits do not comprise inductors.

**Table 4.** Measured integrated inductors at HT.

| Inductor | @ 25 °C (μH) | @ 120 °C (μH) | @ 150 °C (μH) | @ 200 °C (μH) | @ 250 °C (μH) | @ 300 °C (μH) |
|---|---|---|---|---|---|---|
| L1 | 0.99 | 0.99 | 1.03 | 1.09 | 1.26 | 1.36 |
| L2 | 0.975 | 0.975 | 1.1 | 1.105 | 1.31 | 1.3 |
| L3 | 0.86 | 0.86 | 0.855 | 0.9 | 1.05 | 1.14 |
| L4 | 1.5 | 1.28 | 1.3 | 1.4 | 1.5 | 1.6 |
| L5 | 1.45 | 1.45 | 1.45 | 1.5 | 1.58 | 1.58 |
| L6 | 1.7 | 1.7 | 1.7 | 1.7 | 1.8 | 1.9 |

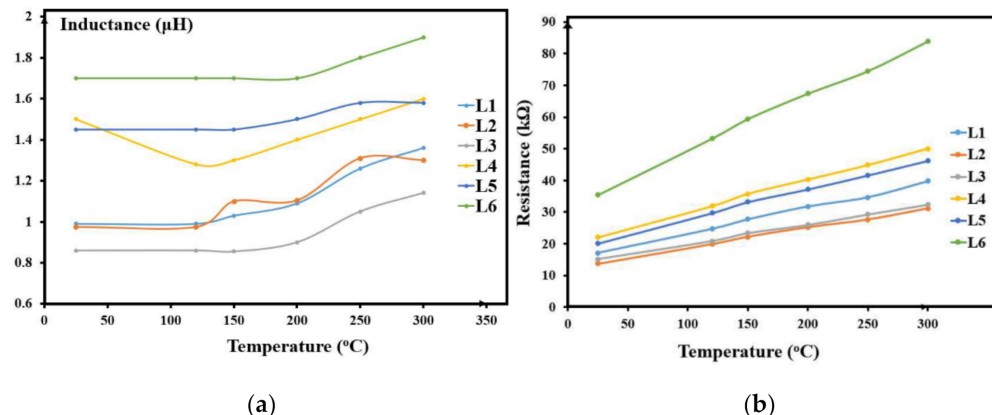

**Figure 6.** HT characterization of integrated inductors: (**a**) inductance value and (**b**) resistance value.

**Table 5.** Measured resistances of integrated inductors at HT.

| Inductor | @ 25 °C (Ω) | @ 120 °C (Ω) | @ 150 °C (Ω) | @ 200 °C (Ω) | @ 250 °C (Ω) | @ 300 °C (Ω) |
|----------|-------------|--------------|--------------|--------------|--------------|--------------|
| L1 | 17.1 | 24.7 | 27.7 | 31.8 | 34.7 | 39.8 |
| L2 | 13.7 | 19.9 | 22.2 | 25.2 | 27.6 | 31.1 |
| L3 | 35.4 | 53.2 | 59.4 | 67.4 | 74.5 | 83.9 |
| L4 | 22.1 | 31.9 | 35.8 | 40.3 | 44.8 | 50 |
| L5 | 20.1 | 29.7 | 33.2 | 37.2 | 41.6 | 46.2 |
| L6 | 15.2 | 20.9 | 23.4 | 26 | 29.3 | 32.4 |

## 3. Circuit Design and Measurements

To validate the functionality of GaN500 technology in high temperature applications such as aerospace and automotive, digital building blocks have been designed and tested. A 4mm × 4mm test chip that was fabricated and tested comprises digital circuits, passive elements, and voltage reference circuits. This chip was designed as a vehicle to validate the functionality of designed circuits at normal and high temperatures. The micrograph of chip is found in Figure 7, and Table 6 summarizes the circuits embedded in that test chip with the corresponding biasing currents and power consumption values.

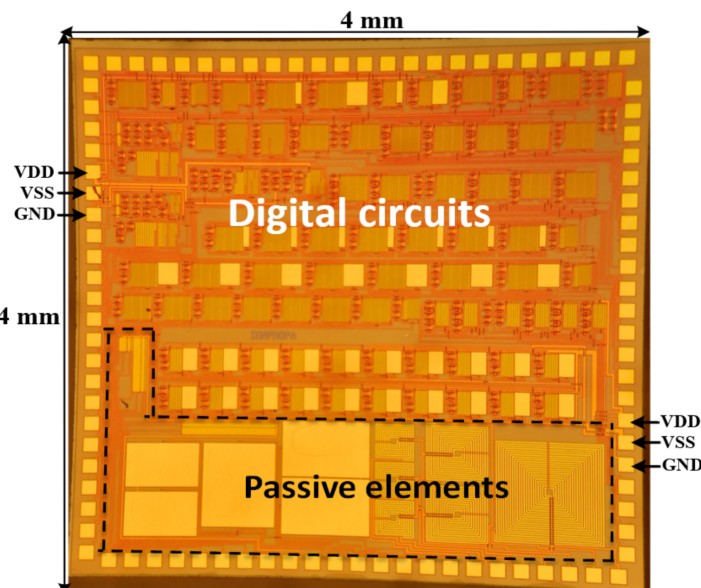

**Figure 7.** Chip micrograph.

**Table 6.** Implemented circuits with corresponding currents and power consumption values.

| Circuit | $I_{DD}$ (mA) | $I_{GND}$ (mA) | $I_{SS}$ (mA) | Power (mW) | Circuit | $I_{DD}$ (mA) | $I_{GND}$ (mA) | $I_{SS}$ (mA) | Power$_{max}$ (mW) |
|---------|--------------|----------------|---------------|------------|---------|--------------|----------------|---------------|--------------------|
| INV1 | <1 | <1 | <1 | 20 | NOR2 | <1 | <1 | <1 | 20 |
| INV2 | 9 | 5 | 4 | 182 | NOR3 | 1.5 | 1 | <1 | 25 |
| INV3 | 6 | 5 | 1 | 98 | VREF1 | <1 | <1 | NA | 0.7 |
| INV4 | <1 | <1 | <1 | 20 | VREF2 | <1 | <1 | NA | 0.6 |
| INV5 | <1 | <1 | <1 | 20 | D Flip-Flop | 6 | 3 | 3 | 126 |
| INV6 | <1 | <1 | <1 | 20 | OSC1 | 13 | 8 | 5 | 252 |
| INV7 | 9 | 5 | 4 | 182 | OSC2 | 13 | 8 | 5 | 252 |
| NAND2 | <1 | <1 | <1 | 20 | DELAY | 150 | 60 | 90 | 3360 |
| NAND3 | <1 | <1 | <1 | 20 | - | - | - | - | - |

The power supply (three terminals) is common for all circuits with VDD = +14 V, VSS = −14 V and 0 V (ground). GaN transistors are normally-on, so it requires negative gate voltage (VGS = −5 V) to turn them off [17]. Therefore, all nominal input (and output) signal swings are between 0 V and −5 V. The chip has two power supplies setting three voltage levels: +14 V, −14 V and 0 V. It was better to add more pads of power supply for stable powering and to control the power of each circuit. However, the pads were limited to control all circuits.

*3.1. Logic Gates: NOT, NAND, NOR*

3.1.1. Inverter (NOT Gate)

This gate is the basic digital circuit prototype implemented with GaN500. The structure schematic of inverter is shown in Figure 8a. Different inverters are implemented by changing the values of the resistors, the transistor size and sometimes by adding a capacitor on the output stage. The input signal (IN) of the inverter controls a GaN driver device. The latter has a drain current limited by R1 to maintain a low power consumption. The remaining part of the circuit performs the role of a voltage level shifter that ensures the voltage level of the output signal (OUT) to be compatible with the input of the next logic circuit stage.

1. INV1: This inverter is designed to have the lowest power consumption. Its transistors have the minimum size specified by the process (L = 500 nm, W = 20 μm) and resistors are very high resistance values (20 kΩ) to limit the current flow as much as possible. In this case, the current in each branch stemming from the power supply is less than 1 mA. The schematics of INV1 is shown in Figure 9a, and its micrograph is presented in Figure 10a. The experimental input-output response at 25 °C and 300 °C of INV1 are shown in Figure 11.

2. INV2: This inverter is designed with a higher power consumption budget (smaller resistance values around 2.5 kΩ). The schematics of INV2 is shown in Figure 9b, and its micrograph is presented in Figure 10b. The experimental input-output response at 25 °C and 300 °C of INV2 are shown in Figure 12.

3. INV3: This inverter is an intermediate solution between the normal inverter (INV2) and the power optimized one (INV1). Its main resistor R1 has a small resistance value (2.5 kΩ), however the level shifter resistors R2 and R3 have a large resistance value (20 kΩ). The schematics of INV3 is shown in Figure 9c, and its micrograph is presented in Figure 10c.

4. INV4: This inverter is used along with INV3 in the digital demodulator system [18]. The schematics of INV4 is shown in Figure 9d, and its micrograph is presented in Figure 10d. Its main goal is to detect and convert a modulated signal with amplitude voltage only higher than −4 V (High-level). Any signal ≤ −4 V (Low-level) will not be detected and will produce a low output level (−5 V). Figure 13 shows how INV4 does not respond at input voltage of −3.5 V (Figure 13a) and starts responding at input voltage of −4 V (Figure 13b). By contrast, INV3 responds in both conditions of

VIN. Both INV3 and INV4 keep working at HT as seen in Figure 14a at 400 °C and Figure 14b after cooling from 450 °C.

5. INV5: This inverter is designed to allow building a 5 MHz ring oscillator (seven stages). A capacitor is added to its output to ensure the required signal delay that allow building a 5 MHz oscillator. The schematics of INV5 is shown in Figure 9e, and its micrograph is presented in Figure 10e. The experimental results of INV5 at 25 °C and 300 °C are shown in Figure 15.

6. INV6: This inverter is designed to allow building a 1 MHz ring oscillator (seven stages). A capacitor is added to its output to produce the stage delay compatible with a 1 MHz oscillator. The schematics of INV6 is shown in Figure 9f, and its micrograph is presented in Figure 10f. The experimental results at 25 °C and 300 °C of INV6 are shown in Figure 16.

7. INV7: This inverter is designed to build the DELAY circuit for high frequency applications (1–5 MHz). A capacitor is added to the output stage to ensure the required delay time. The schematics of INV7 is shown in Figure 9g, and its micrograph is presented in Figure 10g. The experimental results of INV7 at 25 °C and 300 °C are shown in Figure 17.

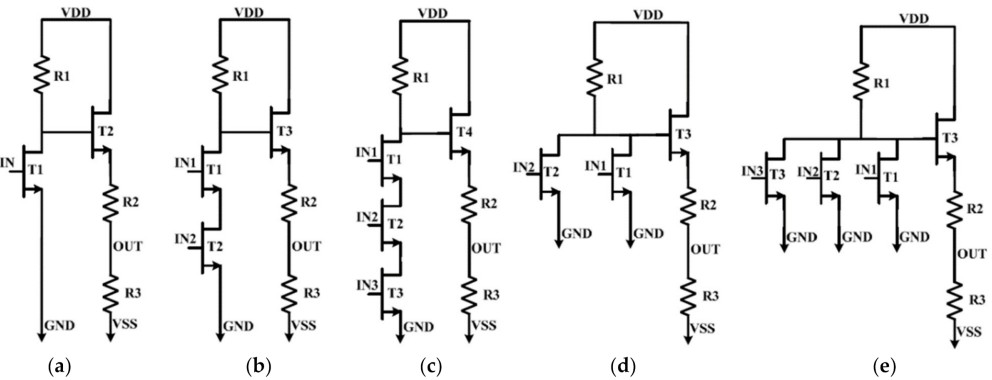

**Figure 8.** Schematics of various implemented logic gates: (**a**) NOT, (**b**) NAND2, (**c**) NAND3, (**d**) NOR2, and (**e**) NOR3.

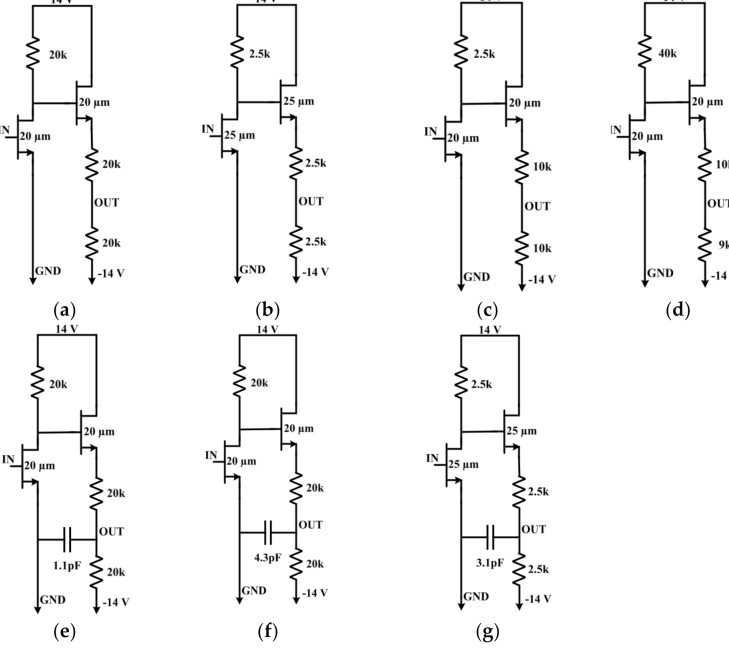

**Figure 9.** Circuit structure of NOT gates: (**a**) INV1, (**b**) INV2, (**c**) INV3, (**d**) INV4, (**e**) INV5, (**f**) INV6, and (**g**) INV7.

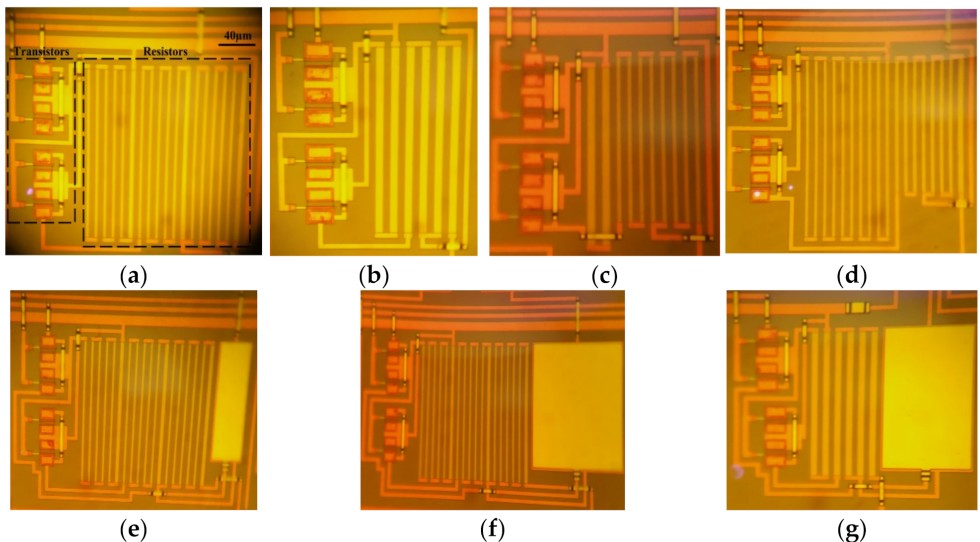

**Figure 10.** Micrographs of various implemented NOT gates: (**a**) INV1, (**b**) INV2, (**c**) INV3, (**d**) INV4, (**e**) INV5, (**f**) INV6, and (**g**) INV7.

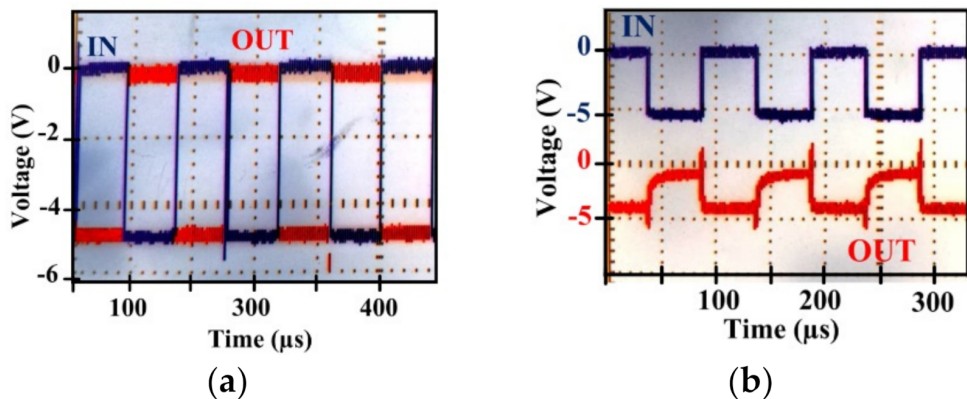

**Figure 11.** Experimental input-output response of INV1: (**a**) at 25 °C and (**b**) at 300 °C.

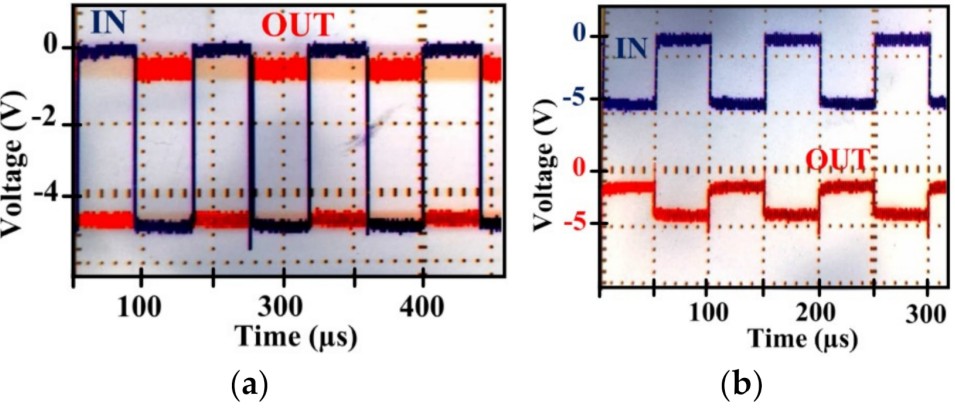

**Figure 12.** Experimental input-output response of INV2: (**a**) at 25 °C and (**b**) at 300 °C.

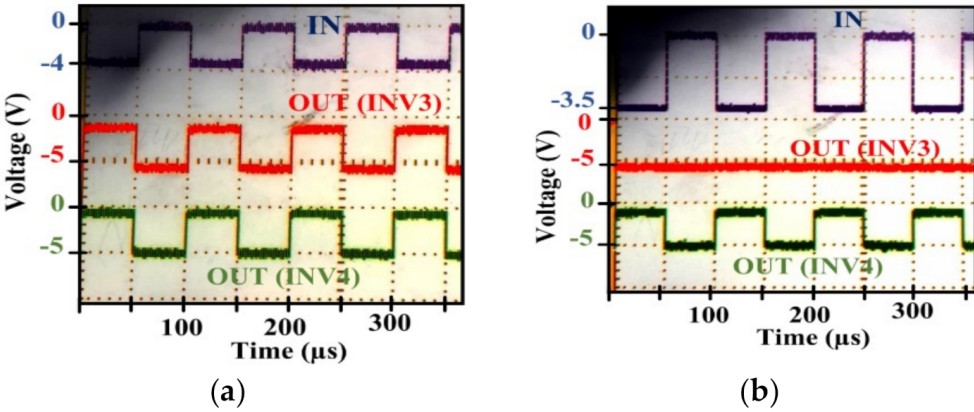

**Figure 13.** Experimental results for INV3 and INV4 at 25 °C with: (**a**) VIN = 0, −4 V and (**b**) VIN = 0, −3.5 V.

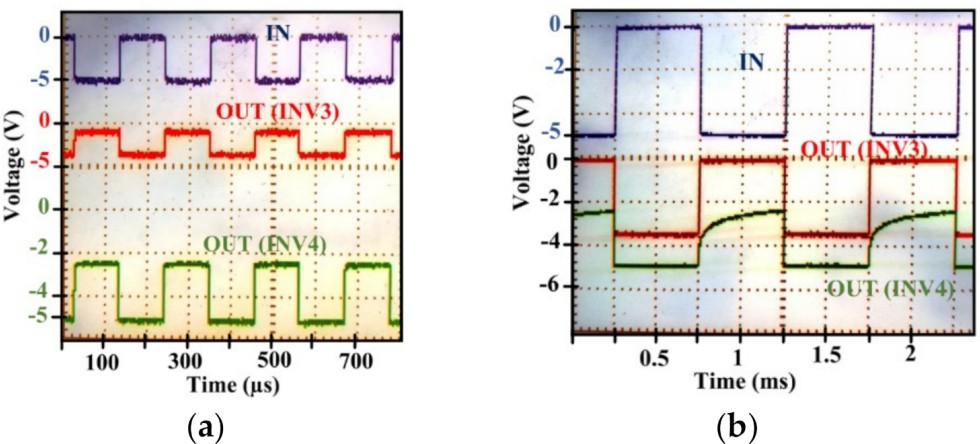

**Figure 14.** Experimental results for INV3 and INV4: (**a**) at 400 °C and (**b**) at 25 °C after heating to 450 °C.

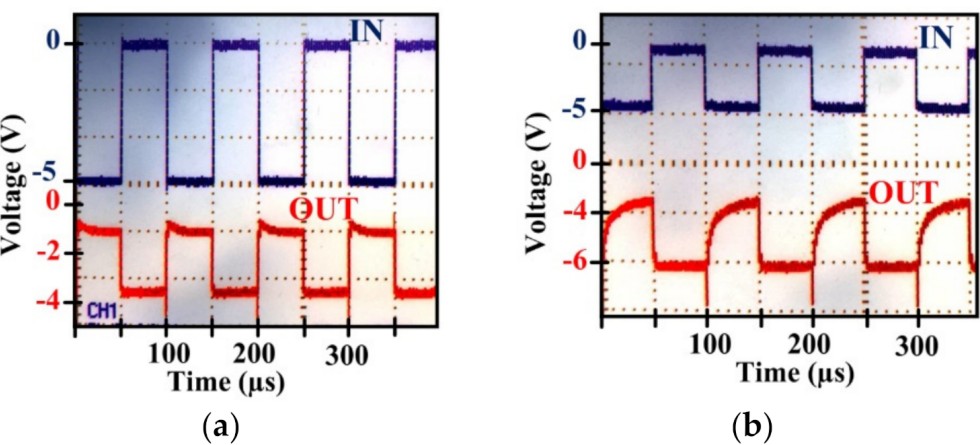

**Figure 15.** Experimental results of INV5: (**a**) at 25 °C and (**b**) at 300 °C.

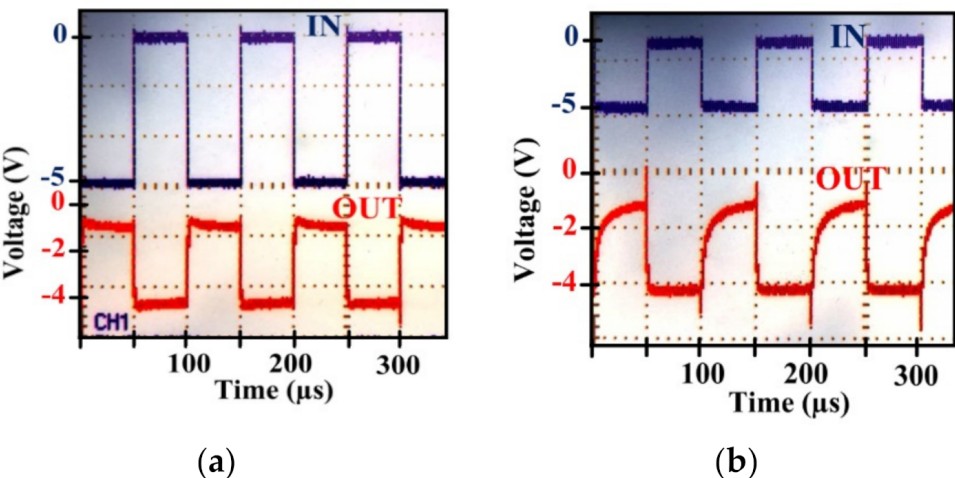

**Figure 16.** Experimental results of INV6: (**a**) at 25 °C and (**b**) at 300 °C.

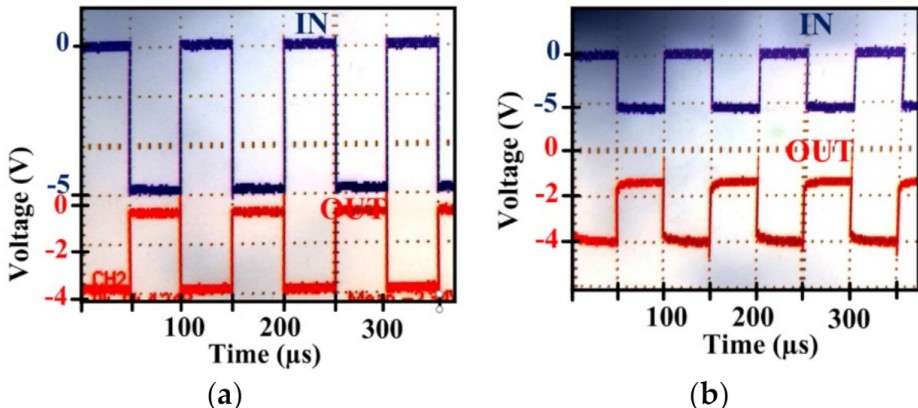

**Figure 17.** Experimental results of INV7: (**a**) at 25 °C and (**b**) at 300 °C.

3.1.2. NAND Gate

The NAND gate is needed to build other circuits such as the D Flip-Flop and the XOR/XNOR. The circuit schematics of a NAND2 and NAND3 gates are shown in Figure 8b,c, respectively. In Figure 8b, two GaN devices in series are used in the driver stage of the NAND2 gate. Similarly, three GaN HEMTs in series are used to form the core of NAND3 gate as shown in Figure 8c. To test the NAND gates, the following stimuli are applied: input logic low = −5 V, input logic high = 0 V, VDD = +14 V and VSS = −14 V. The ±14 V supplies were adopted to ensure stable operation of digital ICs. These values are required to let the level shifters ensure compatibility between the output of one digital circuit and the input of the following digital circuit. The voltage supplies could be reduced to less than ±14 V in case of simple digital ICs (NOT, NAND, NOR), but ±14 V was found necessary for complex digital and mixed ICs. Finding ways to reduce supply voltages and power consumption while maintaining reliable operation over wide temperature ranges was left for future research.

1. NAND2: This gate is designed with optimized power consumption using the smallest transistors and large resistance values. The schematic of NAND2 is shown in Figure 18a, and its micrograph is given in Figure 18c. The transient characteristics of the NAND2 at 25 °C, 300 °C, and 400 °C are shown in Figure 19a–c, respectively. Figure 19d shows the measured response of NAND2 at 25 °C after it was heated to 450 °C. It is of interest that the output responses are different for the various reported experiments. However, these output responses always remain consistent with the respective different input signals that were applied in these experiments. Reproducing

the same stimuli in the various experiment could be done easily with the available test setup.

2.  NAND3: This gate is designed with optimized power consumption using small transistors and large resistance values. The schematics of NAND3 is shown in Figure 18b, and its micrograph is given in Figure 18d. The transient response of the NAND3 is shown in Figure 20 at 25, 170 and 300 °C. In Figure 18b, the third input is connected to IN1 due to the unavailability of a third signal generator at the testing time, however this connection is adequate to test the functionality of the gate.

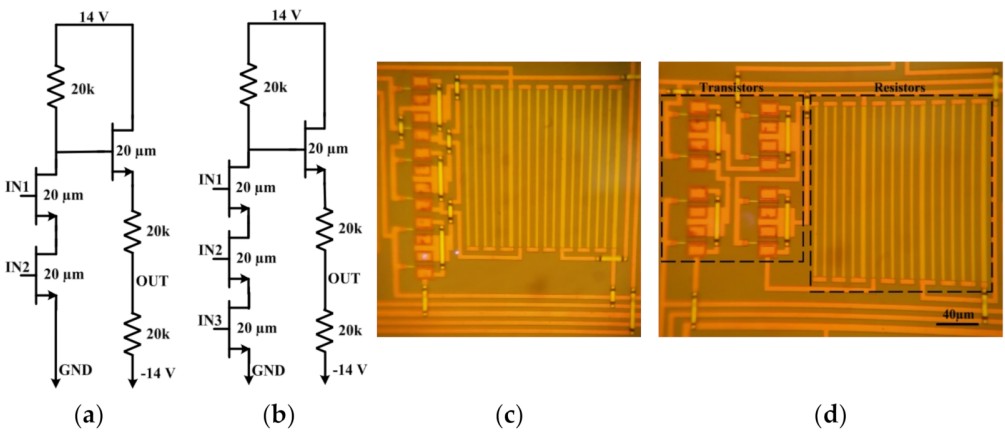

**Figure 18.** Implemented NAND logic gates: (**a**) circuit structure of the NAND2, (**b**) circuit structure of the NAND3, (**c**) micrograph of the NAND2, and (**d**) micrograph of the NAND3.

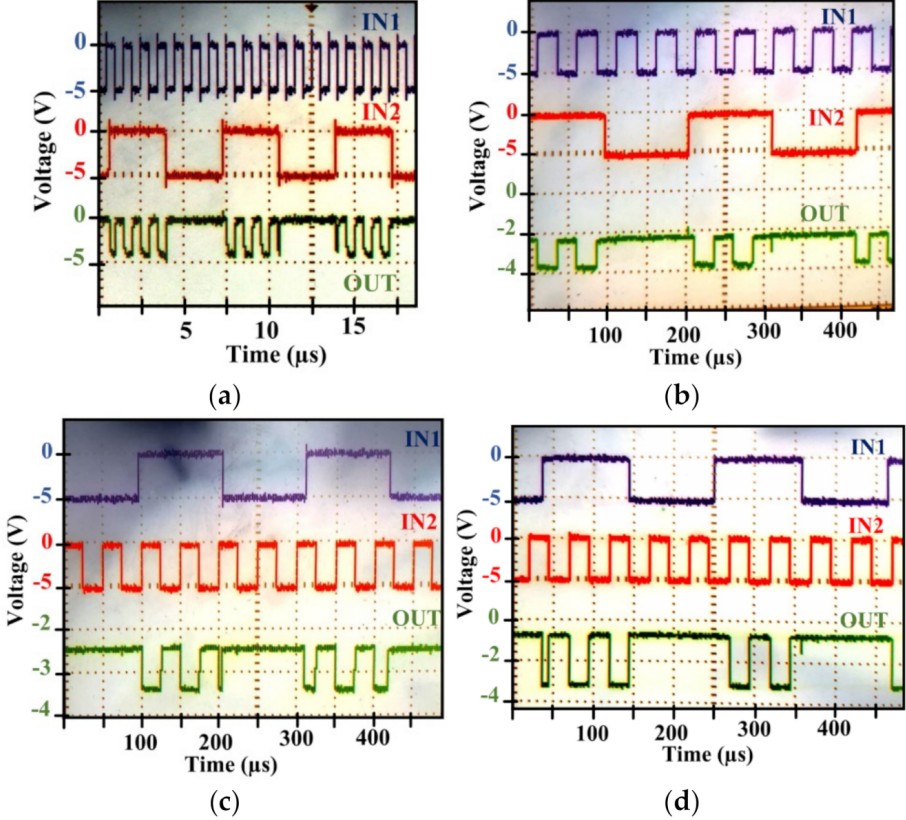

**Figure 19.** Transient response of NAND2 at: (**a**) 25 °C, (**b**) 300 °C, (**c**) 400 °C, and (**d**) 25 °C after heating to 450 °C.

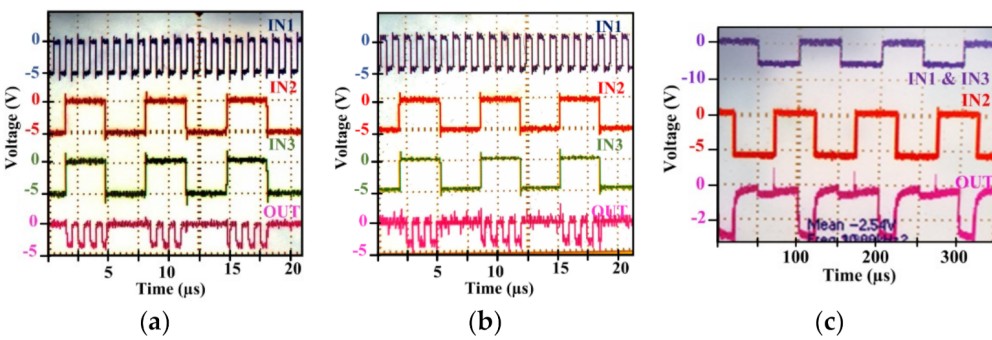

**Figure 20.** Transient response of the NAND3 at: (**a**) 25 °C, (**b**) 170 °C, and (**c**) 300 °C.

### 3.1.3. NOR Gate

The NOR logic gate is designed to ensure that we can use GaN500 technology to implement all basic digital circuits, including the NOR gate. NOR2 and NOR3 gates shown in Figure 8d,e, respectively, used two and three GaN devices in parallel, respectively, in the driver stage.

1. NOR2: this gate is designed with optimized power consumption using the smallest transistors and large resistance values. The schematic of NOR2 is shown in Figure 21a, and its micrograph is given in Figure 21c. The transient response characteristics of NOR2 at 25, 300, and 400 °C are shown in Figure 22a–c, respectively. Figure 22d shows the measured transient response of NOR2 at 25 °C after it was heated to 450 °C.

2. NOR3: This gate is designed with optimized power consumption using the small transistors and large resistance values. The schematics of the NOR3 is shown in Figure 21b, and its micrograph is given in Figure 21d. The transient response characteristics of NOR3 are shown in Figure 23 at 25 and 300 °C. The third input is connected to IN1 due to the unavailability of a third signal generator at the testing time. However, this connection is sufficient to test the functionality of the gate.

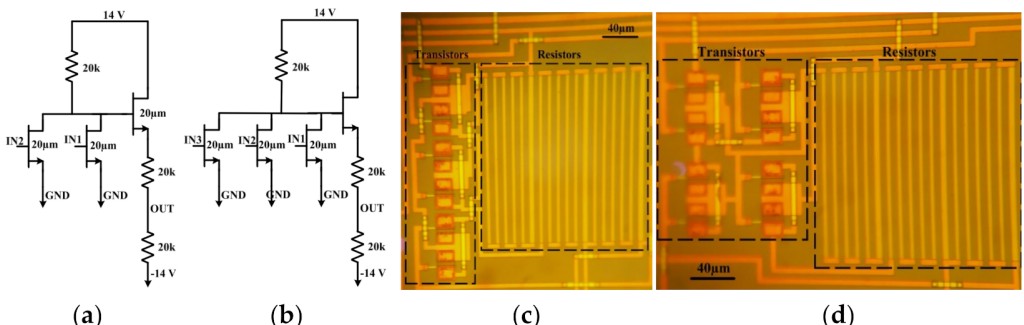

**Figure 21.** Implemented NOR logic gates: (**a**) circuit structure of the NOR2, (**b**) circuit structure of the NOR3, (**c**) micrograph of the NOR2, and (**d**) micrograph of the NOR3.

Unfortunately, we did not perform HT measurements up to 400 °C for all the logic gates (NOT, NAND, NOR) due to certain limitations in our HT test setup, including packaging and number of supported wire bonding connections. In addition, the reliability (MTTF) was also not assessed at this time. This work focused on establishing functionality performance at t = 0 for various HT conditions.

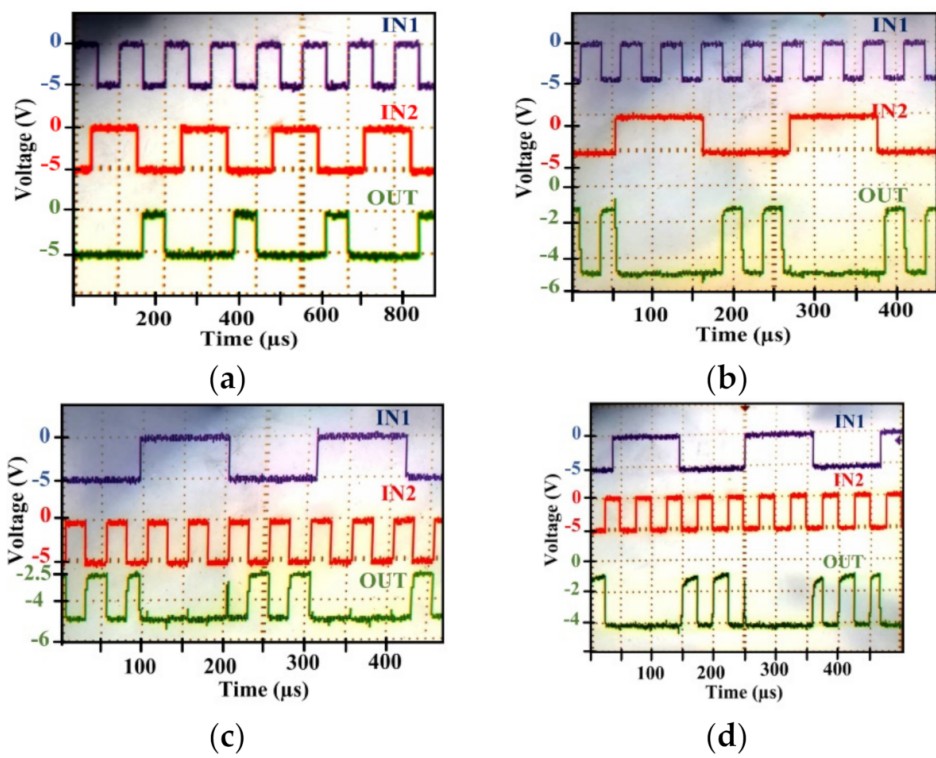

**Figure 22.** Transient response characteristics of the NOR2 gate at: (**a**) 25 °C, (**b**) 300 °C, (**c**) 400 °C, and (**d**) 25 °C after heating to 450 °C.

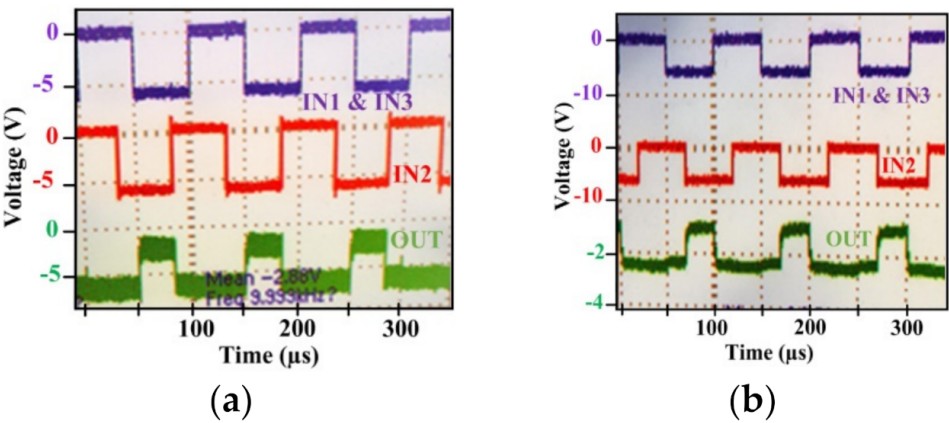

**Figure 23.** Transient response of the NOR3 gate at: (**a**) 25 °C and (**b**) 300 °C.

*3.2. Digital Circuits: Delay, D Flip-Flop, Ring Oscillator*

3.2.1. Delay Circuit

Delay circuit: This circuit is needed in the digital demodulation system [18]. Its schematic diagram is shown in Figure 24a. It is composed of 20 INV7 inverters used to delay the input signal. Five outputs are extracted to provide five different delayed signals. An average delay per stage of 25 ns is obtained. The micrograph is presented in Figure 25a. The tested signal of the fifth output is shown in Figure 26 at 25 and 160 °C. It is clear how the fifth output signal is delayed from the input signal. Beyond 160 °C, the Delay circuit is no longer functional. It is conjectured that the local heating effects due to the high-power dissipation of utilized inverters (182 mW for an INV7 as shown in Table 6, which implies a total dissipation of more than 3.6 W) elevate the junction temperatures above their point of correct functional operation. This implies a significant shift of HS and LS of internal inverters.

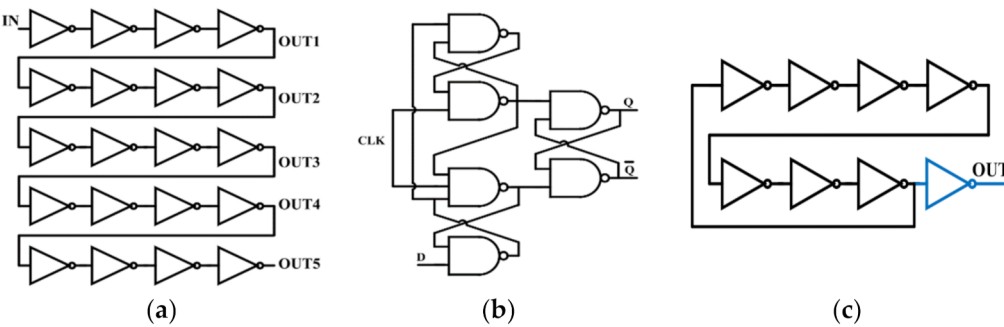

**Figure 24.** Schematics of implemented circuits: (**a**) delay, (**b**) D Flip-Flop, and (**c**) ring oscillator.

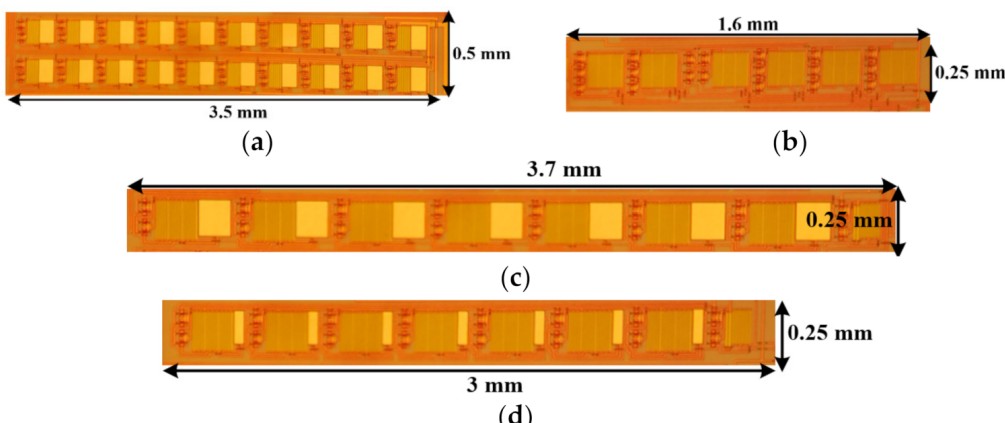

**Figure 25.** Micrographs of: (**a**) delay circuit, (**b**) D Flip-Flop, (**c**) OSC-1, and (**d**) OSC-2.

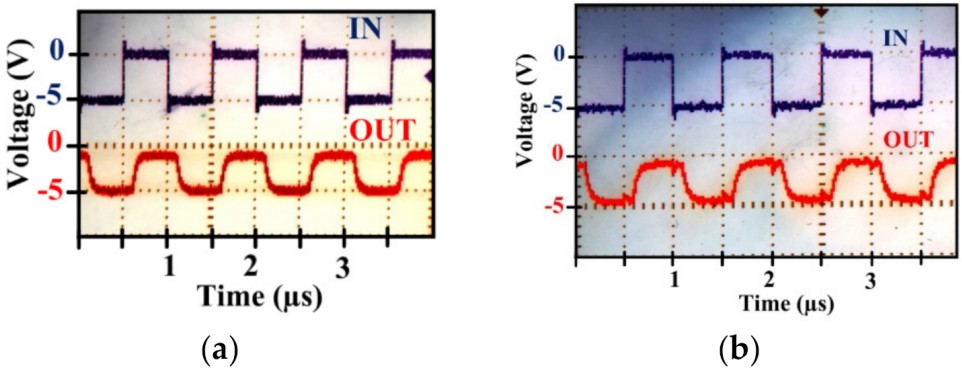

**Figure 26.** Transient response characteristics of the delay-5 cell at: (**a**) 25 °C and (**b**) 160 °C.

### 3.2.2. D Flip-Flop

Figure 24b presents the building block of the D Flip-Flop cell composed of five NAND2 and one NAND3. Although the NANDs show an acceptable functionality at HT beyond 300 °C, the D Flip-Flop does not operate above 160 °C. This is due to shifts of inputs and outputs levels of the NANDs. The micrograph of the D Flip-Flop is presented in Figure 25b and the transient response is shown in Figure 27 at 25, 100, and 160 °C. Similar to the Delay circuit, after 160 °C the D Flip-Flop stops working due to shifts of inputs and outputs levels of the NANDs. More details could be found in [18].

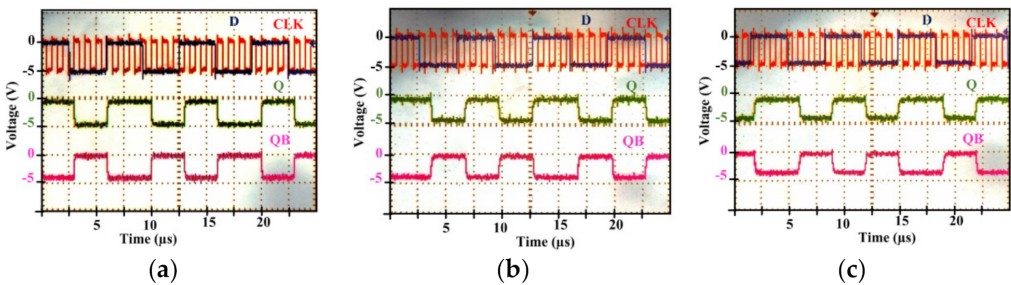

**Figure 27.** Transient response of the D Flip-Flop at: (**a**) 25 °C, (**b**) 100 °C, and (**c**) 160 °C.

### 3.2.3. Ring Oscillator

The ring oscillator circuit is designed to generate the clock signal of modulation and demodulation systems [17]. Different oscillating frequencies are generated to serve different types of applications. The schematic of the ring oscillator is shown in Figure 24c where a ring comprising of seven inverter stages is followed by a final inverter to sharpen the generated waveform.

1.  OSC-1: In this circuit, 7 INV6 stages are used to build a ring oscillator with an oscillation frequency of 1.5 MHz. The micrograph of OSC-1 is depicted in Figure 25c and its transient response is shown in Figure 28a at 25 °C. Figure 28b shows the measured output signal produced by OSC-1 at 25 °C after it was heated to 450 °C.
2.  OSC-2: This oscillator is similar to OSC-1 but it is built using INV5. OSC-2 is designed to generate a 5 MHz oscillation frequency. The micrograph of OSC-2 is presented in Figure 25d and its measured output signal at 25 °C is shown in Figure 29a. Figure 29b shows the measurement of OSC-1 at 25 °C after heating to 450 °C.

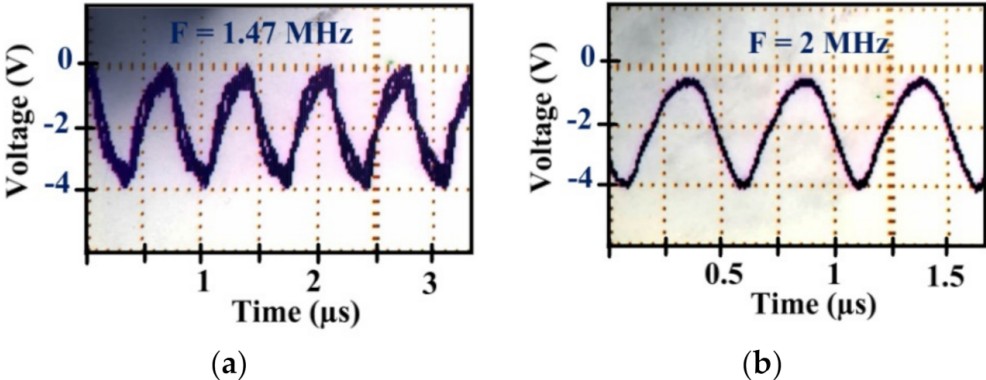

**Figure 28.** Transient response of OSC-1 at: (**a**) 25 °C and (**b**) 25 °C after heating to 450 °C.

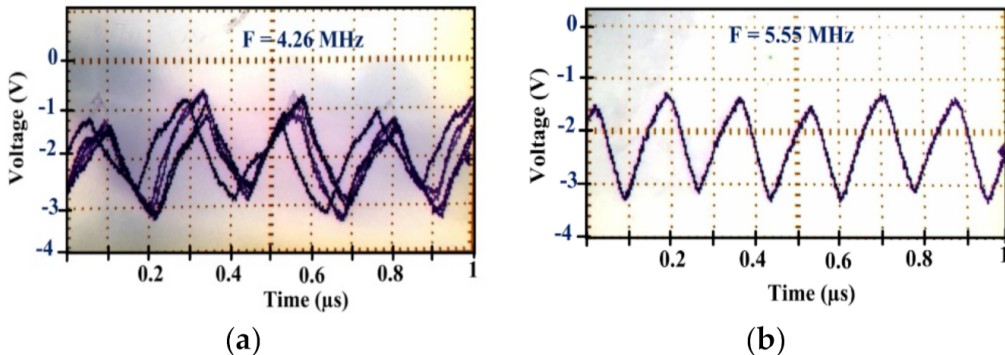

**Figure 29.** Measured output response of OSC-2 at: (**a**) 25 °C and (**b**) 25 °C after heating at 450 °C.

Although the GaN500 transistors [19] and passive elements are still working at HT, up to 600 °C as shown in Section 2, it does not ensure that the logic gates keep working properly at very high temperature. This is more a design stability issue than a components lifetime issue. It is believed that further work would allow optimizing the proposed cells to reach operating temperatures higher than 300 °C in the future. For instance, the test chip reported in this paper could not benefit from the transistor [19] and resistor (Section 2) models that we developed to design circuits more stable at extreme HT because the logic gates were fabricated at the same time as the devices from which improved device models were developed.

Moreover, the standard PDK for this GaN500 technology/process flow uses a gate anneal temperature that is much lower than the measurement temperatures explored in this work, e.g., 500 °C and higher. This resulted in changes to the HEMT transfer characteristics due to the HT testing itself, which shifted the optimal operating points. For future work, it emphasizes the need to modify the PDK once the experimental design processes (e.g., gate annealing temperature at or above the intended operating temperature) are completed, and re-testing and new modeling are performed.

### 3.3. Voltage Reference Circuit

Another circuit embedded on the same chip is a voltage reference. This first reported voltage reference implemented with GaN devices can operate up to 550 °C. It is built with three GaN500 devices and two resistors, as shown in Figure 30. The idea behind the proposed voltage reference is based on using the GaN transistor in its saturation region (above VDS = 5 V) as a constant current source. The latter feeds a resistive load to generate a constant voltage. When the temperature increases, the decreasing current passing through the conductive channel of the GaN transistor is compensated by the increasing resistance of load resistors. Three GaN devices are connected in series to improve the stability of the generated saturation current and to reduce its value. The load resistors are adapted to give the desired voltage reference.

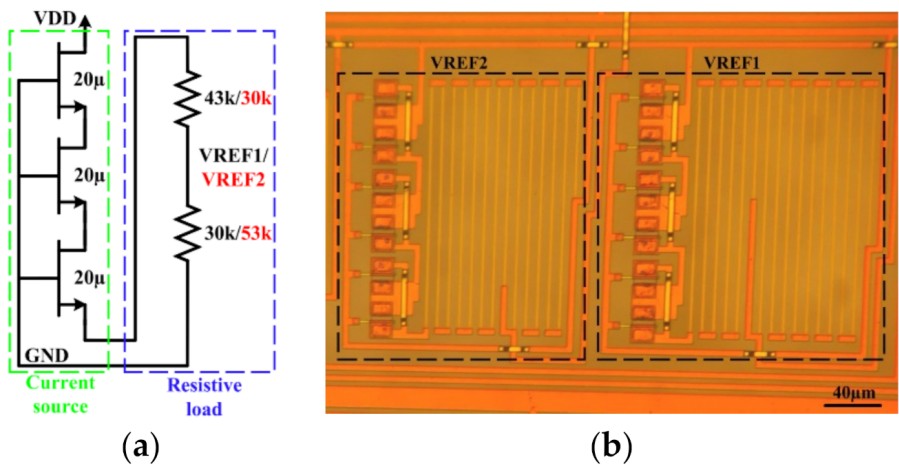

**Figure 30.** Integrated voltage references: (**a**) schematic design and (**b**) micrograph.

Two voltage references are required to operate other systems not mentioned in this work [16] with VREF1 = 3.3 V and VREF2 = 2.3 V. As a reference circuit, when the supply voltage increases, the output voltage gradually increases to become stable after the supply achieves a certain threshold. Then, the output voltage stays constant over the remaining range of supply voltages at a specific temperature. The experimental measurements shown in Figure 31 show the outputs of the voltage references observed when sweeping the supply voltage from 0 to 16 V over a wide temperature range between 25 and 600 °C. The results show that the reference voltage is relatively stable over the full temperature range until 550 °C, at supply voltages above 4 V.

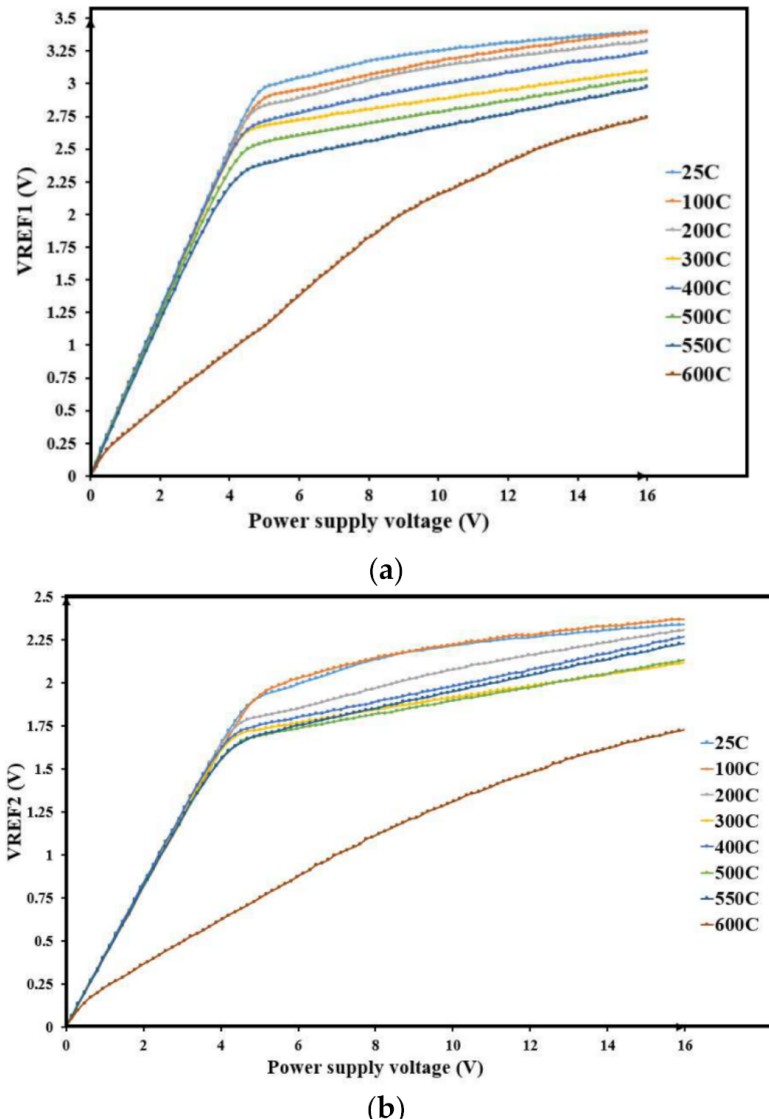

**Figure 31.** Supply voltage dependence of the proposed voltage references in the temperature range 25–600 °C: (**a**) VREF1 and (**b**) VREF2.

Figure 32 shows the outputs of VREF1 and VREF2 at a nominal supply voltage (14 V) over a wide range of temperature 25–600 °C. These references exhibit temperature coefficients TC1 = 293 ppm/°C and TC2 = 242 ppm/°C. Although the reported TCs do not compete with silicon-based bandgap references, the range of operating temperatures is much higher. Indeed, the voltage references reported here are the first integrated voltage references operating at such extreme temperatures.

The GaN500 HT ICs reported in the work can be compared with various recently published work implementing HT ICs based on GaN devices. Table 7 summarizes the type of ICs that have been reported in the literature and their maximum operating temperature. As noticed before, most ICs were not reported to work at temperatures exceeding 300 °C, except in [21] where the implemented differential amplifier and inverter showed functionality at 500 °C. However, the employed GaN technology in [21] was never offered commercially and no further development or publication has been reported since 2015.

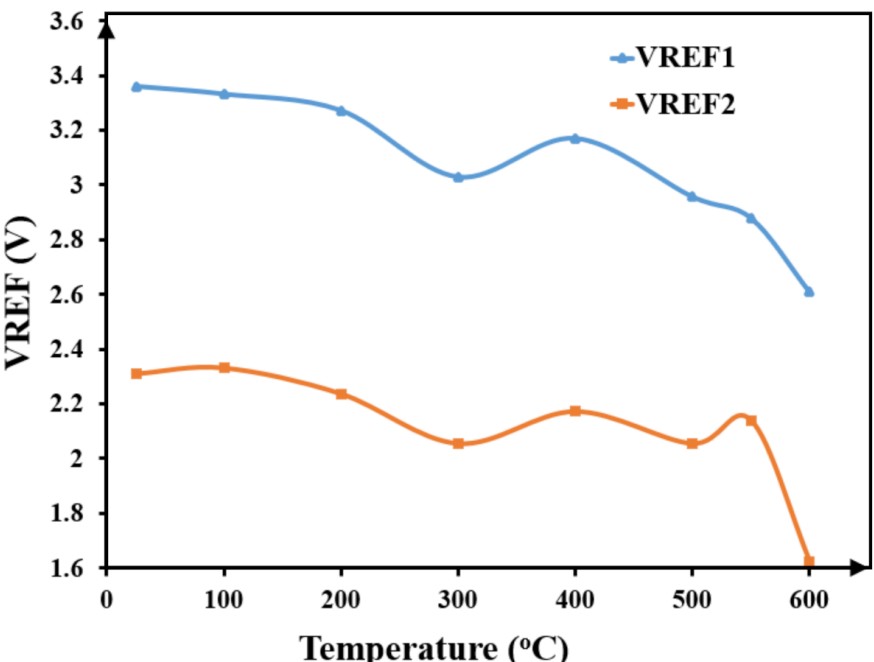

**Figure 32.** Experimental voltage reference outputs in the temperature range of 25–600 °C at supply voltage = 14 V.

**Table 7.** Maximum operating temperature of previously reported GaN ICs.

| Reference | GaN Technology | Reported ICs | Temperature | Year |
|---|---|---|---|---|
| [10] | GaN HEMT | Oscillator | 250 °C | 2013 |
| [11] | GaN MOS-FET/HEMT | Inverter | 300 °C | 2014 |
| [21] | GaN HEMT | Inverter, Dif. Amp | 500 °C | 2015 |
| [22] | GaN HEMT | PWM | 250 °C | 2015 |
| [12] | GaN HEMT | Inverter, Oscillator | 200 °C | 2017 |
| [23] | GaN MIS-HFET | Gate driver | 250 °C | 2019 |
| [14] | GaN HEMT | Gate drive, Level shifter | 200 °C | 2019 |
| [15] | GaN MOSFET | Inverter | 300 °C | 2020 |
| This work | GaN HEMT | NAND, NOR, NOT, D Flip-Flop *, Delay *, Oscillator [1], Voltage reference [+] | 400 °C, 160 °C *, 550 °C [+] | 2021 |

[1] Tested at 25 °C after heating to 450 °C. * and [+] are used to clarify the maximum operation temperature of each circuit.

## 4. Conclusions

This paper reported the implementation and validation of several digital integrated circuits that were successfully fabricated using the GaN500 technology provided by the National Research Council of Canada (NRC). The electrical characteristics of integrated passive elements were tested at various temperatures ranging from 25 °C to 600 °C. In addition, we experimentally demonstrated digital logic gates and voltage reference ICs implemented using GaN500. The prototype ICs were characterized at temperatures ranging from 25 °C to 600 °C. Experimentally tested NOT, NOR and NAND logic gates were shown to have very stable characteristics at temperatures up to 400 °C. In addition, fundamental

high temperature building blocks were implemented based on the reported GaN logic ICs, including D Flip-Flops, ring oscillators, and delay circuits. Moreover, a proposed voltage reference circuit was fabricated and experimentally validated at temperatures as high as 550 °C. The designed voltage reference is rather basic and would need further improvements. Nevertheless, the reported circuit is a significant first step towards a stable voltage reference operating at very high temperatures exceeding 500 °C.

In this work, we aimed to test the functionality of the first GaN500 based digital circuits and to extract accurate models of passive elements operating at HT. The next future work step is to use the extracted models to improve the design of analog and digital circuits to enable the design of more stable circuits operating at temperatures exceeding 500 °C. The reported work is proposed as a contribution calling for further improvements and developments of GaN-based ICs, especially for GaN-based wireless monitoring systems targeting high-temperature harsh environments.

**Author Contributions:** Conceptualization, A.H.; methodology, A.H.; validation, A.H.; formal analysis, A.H.; investigation, A.H.; data collection, A.H.; writing—original draft preparation, A.H.; writing—review and editing, A.H., J.-P.N., Y.S. and M.S.; visualization, A.H.; supervision, Y.S. and M.S.; funding acquisition, Y.S. and M.S; design layout reviews, full-wafer GaN MMIC fabrication and in-process I-V testing, manuscript review and editing, J.-P.N. All authors have read and agreed to the published version of the manuscript.

**Funding:** This work was supported in part by the Natural Sciences and Engineering Research Council of Canada (STPGP463394), CMC Microsystems, National Research Council Canada (NRC), Safran and Airbus Defence and Space.

**Conflicts of Interest:** The authors declare no conflict of interest.

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
