# Peer review of "Circuit Techniques in GaN Technology for High-Temperature Environments"

_electronics, doi:10.3390/electronics11010042_

Round 1
Reviewer 1 Report
I think this topic is very interesting and the authors did very nice work.
One comment: The authors need to clarify the new contributions of the paper, maybe elucidate the current status of the literature study, and compared to the literature study, what's the new improvement or new findings in the paper.
Thanks.
Author Response
Manuscript ID: electronics-1490282
Paper Title: Circuit Techniques in GaN technology for High-Temperature Environments
Authors: Ahmad Hassan *, Jean-Paul Noël, Yvon Savaria, Mohamad Sawan
Authors would like to thank the editors and reviewers for providing valuable comments to improve our manuscript. We have done the modifications as suggested and following are the answers, in red color, with excerpts from the revised version of our paper.
Response to Reviewer 1 Comments
Reviewer # |
|
Comment 1) |
I think this topic is very interesting and the authors did very nice work. One comment: The authors need to clarify the new contributions of the paper, maybe elucidate the current status of the literature study, and compared to the literature study, what's the new improvement or new findings in the paper |
Authors |
Thank you for your appreciated feedback and comment. The main contribution of the paper is to demonstrate the effectiveness of commercially available GaN technology (GaN500) to implement high temperature digital ICs. For further clarification, the following is added to the introduction: “However, despite the development and characterization of several GaN devices operating at extreme temperatures exceeding 400 oC [7], 600 oC [8] and 800 oC [9], the implemented ICs based on these devices are very limited in circuit complexity level and achieved HT. This includes oscillator, inverter, and level shifter circuits with maximum operating temperatures up to 300 oC [10-14].” In addition, a comparison table (Table 7) is added to the new version of the paper in section 3.2 as follows: “The GaN500 HT ICs reported in the work are compared with various recently published work implementing HT ICs based on GaN devices, Table 7 summarizes the type of implemented ICs and the maximum operating temperature of each circuit. As noticed before, most ICs were not reported to work at temperatures exceeding 300 oC, except in [22] where the implemented differential amplifier and inverter showed functionality at 500 oC. However, the employed GaN technology in [22] was never offered commercially and no further development or publication has been reported since 2015. ” |

Reviewer 2 Report
Editorial notices:
In the manuscript authors present results of the design and characterisation of high temperature passive and active electronic devices based on GaN on SiC technology. Resistors, capacitors, inductors as well as digital integrated circuits, such as logical gates, flip-flop, delay line and ring oscillator were electrically characterized in elevated temperatures. Scientific investigation and presented data are interesting, however some remarks must be done:
- Unfortunately, there is no comparison to other HT devices and circuit made in other technologies. In my opinion this is the weakest point of this manuscript. Limited number of foreign references also reflect this. Seven of twelve references are self-citations. Consequently, the comparison with other results is very limited or not present.
- A coefficient typically used in the field of electronics that describes the temperature dependence of a resistance is TC (Temperature Coefficient), why authors avoid this parameter?
- Please explain the “Power” in table 6 – is it the static or dynamic power consumption?
- The symbol of transistor in Figure 8 are different from others presented in Figures 9, 18 and 21. Please unify them.
- 11, line 268 – “The experimental and simulation results (…) are shown in Figure 17” – however there is no visible results of simulation in Figure 17. Please comment it.
- 15, line 344 – “All presented logic gates (…)are functional at 400C and higher temperature.” – this sentence isn’t supported by presented data. Electrical characterization of NAND3 and NOR3- was made, or at least presented, up to 300C. Please comment it or correct it.
- All oscilloscope graphs are presented as the photos of the oscilloscope screen. Most modern digital devices have the ability to acquire the plots directly to the computer. Presented photos have different sizes and proportions. If possible, please improve the quality of figures.
- Descriptions in Figures 3, 4, 5, 6, 31and 32 are tiny, please enlarge it.
Author Response
Manuscript ID: electronics-1490282
Paper Title: Circuit Techniques in GaN technology for High-Temperature Environments
Authors: Ahmad Hassan *, Jean-Paul Noël, Yvon Savaria, Mohamad Sawan
Authors would like to thank the editors and reviewers for providing valuable comments to improve our manuscript. We have done the modifications as suggested and following are the answers, in red color, with excerpts from the revised version of our paper.
Response to Reviewer 2 Comments
Reviewer #2 |
|
Comment 1) |
Unfortunately, there is no comparison to other HT devices and circuit made in other technologies. In my opinion this is the weakest point of this manuscript. Limited number of foreign references also reflect this. Seven of twelve references are self-citations. Consequently, the comparison with other results is very limited or not present. |
Authors |
Thank you for your important comment. A comparison table (Table 7) of our work with 8 references in the literature is added to the new version of the paper in section 3.2 as follows: “The GaN500 HT ICs reported in the work are compared with various recently published work implementing HT ICs based on GaN devices, Table 7 summarizes the type of implemented ICs and the maximum operating temperature of each circuit. As noticed before, most ICs were not reported to work at temperatures exceeding 300 oC, except in [22] where the implemented differential amplifier and inverter showed functionality at 500 oC. However, the employed GaN technology in [22] was never offered commercially and no further development or publication has been reported since 2015. ” In addition, the following is added to the introduction: “However, despite the development and characterization of several GaN devices operating at extreme temperatures exceeding 400 oC [7], 600 oC [8] and 800 oC [9], the implemented ICs based on these devices are very limited in circuit complexity level and achieved HT. This includes oscillator, inverter, and level shifter circuits with maximum operating temperatures up to 300 oC [10-14].” |
Comment 2) |
A coefficient typically used in the field of electronics that describes the temperature dependence of a resistance is TC (Temperature Coefficient), why authors avoid this parameter? |
Authors |
TC is usually used to present the change of a certain stable parameter with temperature. In section 3.2 (voltage reference circuit), we use TC to show the stability of constant voltage reference with temperature. However, we didn’t use TC with the characterized resistors because we are not claiming constant and stable resistors. |
Comment 3) |
Please explain the “Power” in table 6 – is it the static or dynamic power consumption? |
Authors |
The “Power” in Table 6 is the maximum power including static and dynamic power consumption. “Powermax” is added to table 6. |
Comment 4) |
The symbol of transistor in Figure 8 are different from others presented in Figures 9, 18 and 21. Please unify them. |
Authors |
Correction done in figures 8, 18 and 21. |
Comment 5) |
11, line 268 – “The experimental and simulation results (…) are shown in Figure 17” – however there is no visible results of simulation in Figure 17. Please comment it. |
Authors |
Correction is done in the new version as follows: “The experimental results at 25 oC and 300 oC of INV7 are shown in Figure 17.” |
Comment 6) |
15, line 344 – “All presented logic gates (…)are functional at 400C and higher temperature.” – this sentence isn’t supported by presented data. Electrical characterization of NAND3 and NOR3- was made, or at least presented, up to 300C. Please comment it or correct it. |
Authors |
The correction is done to remove confusion as follows: “Unfortunately, we did not perform HT measurements up to 400 oC for all the logic gates (NOT, NAND, NOR) due to certain limitations in our HT test setup, including packaging and number of supported wire bonding connections. In addition, the reliability (MTTF) was also not assessed at this time. This work focused on establishing functionality performance at t = 0 for any given HT.” |
Comment 7) |
All oscilloscope graphs are presented as the photos of the oscilloscope screen. Most modern digital devices have the ability to acquire the plots directly to the computer. Presented photos have different sizes and proportions. If possible, please improve the quality of figures. |
Authors |
Sorry for the low quality of figures. The results were taken by camera as quick reporting of results. However, we couldn’t repeat later the measurements to take screenshots from the oscilloscope due to access restrictions to the high temperature testing room. |
Comment 8) |
Descriptions in Figures 3, 4, 5, 6, 31and 32 are tiny, please enlarge it. |
Authors |
Done. |

Round 2
Reviewer 2 Report
Dear Authors, thank you for taking into account all comments. Now, in my opinion, manuscript can be published after editor remarks.
Author Response
Thank you for your appreciated feedback.